# Computational Mass of Living Matter: A Free-Energy and Partition-Theoretic Model of Consciousness

## Abstract

We introduce a dimensionless construct—the *computational mass* of living matter—that links combinatorial internal complexity with adaptive inference under the free energy principle (FEP). The framework combines: (i) an extended mass–energy–information balance for living systems (motivated by a prior "living mass" decomposition), (ii) Markov-blanket and variational-free-energy formulations of self-organization and active inference, and (iii) Hardy–Ramanujan asymptotics together with Rademacher's exact series for the integer partition function $p(n)$. The discrete index $n \in \mathbb{N}$ is defined as a dimensionless complexity parameter derived from an energy (or information) scale, so that $p(n)$ provides a mathematically controlled proxy for the growth of admissible internal configurations. We define

$$M_{\text{comp}}(n) = \Psi \, p(n) \, \exp\left(-F_{\text{eff}}(n)\right),$$

where $F_{\text{eff}}(n)$ is an effective (time-averaged, normalized) variational free energy and $\Psi$ is a normalization constant. The partition term measures structural capacity; the exponential term penalizes poorly adapted, high-surprisal regimes as formalized by FEP. Using the asymptotics of $\log \, p(n)$, we characterize monotonicity regions and critical regimes where marginal combinatorial gains are outweighed by marginal free-energy costs, yielding a phase-transition–like criterion for an operational notion of computational consciousness (efficient self-modeling under active inference). We also give rigorous upper/lower bounds on $M_{\text{comp}}(n)$ and show how the definition specializes to variational generative AI models.

**Keywords:** free energy principle, active inference, Hardy–Ramanujan; Rademacher series, computational consciousness, AGI

## 1. Introduction

The free energy principle (FEP) proposes that adaptive biological systems can be understood as minimizing a variational free-energy functional that upper-bounds surprisal, thereby supporting perception, action, and learning within a unified formalism. The Markov blanket formalism provides the statistical boundary conditions under which internal states and external states interact only through sensory and active states, enabling an active-inference interpretation of self-organization.

A recurring gap in both neuroscience theory and AI theory is a compact mathematical quantity that simultaneously (a) captures the *combinatorial capacity* of a system's internal configurations and (b) captures the *selective pressure* exerted by adaptation and prediction. The present paper proposes such a

quantity by combining (i) analytic number theory for configuration growth (integer partitions) with (ii) the variational mechanics of FEP/active inference.

We take as a motivating starting point a "living mass" decomposition in which the rest energy $m_{\text{living}}c^2$ is treated as comprising internal, structural, and informational components, including thermodynamic and information-theoretic terms (e.g., Gibbs free energies, configurational contributions, and a DNA/information functional). Our contribution is to (1) formalize a dimensionless complexity index $n$, (2) use partition theory to model the growth of admissible internal configurations, and (3) couple that growth to effective variational free energy to obtain a tractable, analyzable "computational mass" $M_{\text{comp}}(n)$.

## 2. Mathematical preliminaries

### 2.1. Variational free energy and Markov blankets

In FEP-based accounts, variational free energy is a functional of an approximate posterior (recognition density) and acts as an upper bound on surprisal, so minimizing free energy reduces an upper bound on $-\log\ p(o)$ for observations $o$. Under the Markov blanket condition, external and internal states are conditionally independent given blanket states (sensory and active), and this conditional-independence structure supports an active-inference reading of self-evidencing dynamics.

We will treat $F(t)$ as a dimensionless quantity measured in nats (natural logarithms) and define a normalized time-averaged free energy $F_{\text{eff}}(n)$ for systems indexed by complexity $n$. When one wants a physical-energy unit, one can multiply dimensionless free energy by a scale such as $k_B T$, but the model below remains dimensionless by construction.

### 2.2. Integer partitions: Hardy–Ramanujan and Rademacher

Let $p(n)$ denote the number of partitions of the integer $n$. The Hardy–Ramanujan circle-method analysis yields a celebrated asymptotic growth formula for $p(n)$, showing that $p(n)$ grows roughly like $\exp\left(\Theta(\sqrt{n})\right)$. Rademacher refined this analysis to obtain an absolutely convergent exact series representation for $p(n)$, turning the asymptotic into an exact formula suitable for high-accuracy computation via truncation.

We will use two properties: (i) $\log\ p(n)$ has leading growth proportional to $\sqrt{n}$, and (ii) Rademacher exactness provides a principled numerical route for computing $p(n)$ when desired.

### 2.3. Dimensionless complexity index $n$

To connect physical constraints to a discrete combinatorial parameter, we introduce a dimensionless complexity index based on an energy (or information) scale.

**Lemma 1 (Dimensionless complexity parameter)**

Let $E_{\text{tot}}$ be a total effective internal energy budget available to sustain distinguishable functional states (not necessarily total rest energy), and let $\epsilon_0 > 0$ be a characteristic "resolution" or quantum of effective energy that separates distinguishable internal configurations. Define

$$n := \left\lfloor \frac{E_{\text{tot}}}{\epsilon_0} \right\rfloor \in \mathbb{N}.$$

Then $n$ is dimensionless and invariant under unit rescaling of energy.

*Comment:* In biological systems, $\epsilon_0$ may be taken as a thermodynamic scale (e.g., proportional to $k_B T$) or a task-relevant minimal free-energy difference between reliably distinct internal states; in AI systems, $\epsilon_0$ can be an effective resolution in log-likelihood or energy-based model units.

**Remark 1 (Information-based alternative)**

If an internal state random variable $X$ has entropy $H(X)$ and a reference entropy quantum $H_0$ (e.g., 1 nat or 1 bit), one can define $n_{\text{info}} := \lfloor H(X)/H_0 \rfloor$ as an alternative dimensionless index. This paper proceeds with the energy-normalized $n$ for clearer coupling to physical constraints.

## 2.4. Why partitions appear (proxy justification)

We require a controlled proxy for "how many ways internal resources can be structured." Integer partitions provide an analytically tractable, rigorously understood configuration-counting function.[3]

**Proposition 1 (Partitions as an exchangeable configuration proxy)**

Suppose $n$ effective quanta are distributable across internal modes and only the multiset of allocations matters (exchangeability across modes). Then the number of distinct allocation types is upper-bounded (and in common idealizations approximated) by an integer-partition count $p(n)$, with exactness in the limit where mode identities do not carry additional labels.

*Proof sketch:* Partition generating functions encode unlabeled allocations, and their coefficients yield the number of allocation types; the unrestricted limit corresponds to the classical partition generating function whose coefficients are $p(n)$. This proposition does not claim that neurobiology "is" partition theory; it provides a conservative and mathematically explicit surrogate for generic combinatorial explosion of internal configuration types.

## 3. From living mass to computational mass model

### 3.1. Motivating "living mass" decomposition

A motivating decomposition (from the provided source material) treats living rest energy as including internal, structural, and informational contributions, expressed schematically as

$$m_{\text{living}}c^2 = E_{\text{internal}} + E_{\text{structural}} + E_{\text{informational}},$$

with possible expansions in terms of Gibbs free energies, configurational terms, and an information functional associated with biological codes (e.g., DNA). We adopt this decomposition as a *motivation* for treating information-bearing structure as a first-class contributor to the "effective capacity" of living matter, while keeping the main construct below dimensionless and not directly equated to kilograms.

## 3.2. Effective free energy at complexity $n$

For each complexity $n$, assume the system admits a variational free-energy trajectory $F_n(t)$ under an active-inference dynamics consistent with Markov blanket constraints. Define the normalized effective free energy

$$F_{\text{eff}}(n) := \frac{1}{Z_n} \int_0^{T_n} F_n(t)\, dt,$$

where $T_n$ is a task- or lifetime-relevant horizon and $Z_n > 0$ normalizes units (e.g., to keep $F_{\text{eff}}$ dimensionless and comparable across $n$).

## 3.3. Definition and normalization of computational mass

We now define the central quantity.

**Definition 1 (Computational mass)**
Let $n \in \mathbb{N}$ be a dimensionless complexity index, let $p(n)$ be the partition function, and let $F_{\text{eff}}(n) \geq 0$ be the effective normalized variational free energy. Define

$$M_{\text{comp}}(n) := \Psi\, p(n)\, \exp\left(-F_{\text{eff}}(n)\right),$$

where $\Psi > 0$ is dimensionless.

Here $p(n)$ is the structural capacity proxy (configuration-type count) and $\exp\left(-F_{\text{eff}}\right)$ is an adaptation filter derived from the free-energy bound on surprisal.

**Normalization (fixing $\Psi$)**
Choose a reference complexity $n_{\text{ref}}$ (e.g., a minimally viable system at the modeling scale) and impose $M_{\text{comp}}(n_{\text{ref}}) = 1$. Then

$$\Psi := \frac{1}{p(n_{\text{ref}})\exp\left(-F_{\text{eff}}(n_{\text{ref}})\right)}.$$

Under this convention, $M_{\text{comp}}(n)$ is a dimensionless *ratio* relative to the reference system.

## 3.4. Monotonicity and critical points

Define the *combinatorial efficiency* function

$$\eta(n) := \frac{d}{dn} \log\, p(n) - \frac{d}{dn} F_{\text{eff}}(n).$$

**Theorem 1 (Monotonicity regions)**

Assume $F_{\text{eff}}(n)$ is differentiable on an interval $I \subset \mathbb{R}_+$ and extend $p(n)$ to a smooth asymptotic surrogate for analysis. Then:

1. If $\eta(n) > 0$ on $I$, $M_{\text{comp}}(n)$ is strictly increasing on $I$.

2. If $\eta(n) < 0$ on $I$, $M_{\text{comp}}(n)$ is strictly decreasing on $I$.

3. Interior critical points satisfy $\eta(n) = 0$, and local maxima occur where additionally $\eta'(n) < 0$.

*Proof.* Since $\frac{d}{dn} \log\, M_{\text{comp}}(n) = \eta(n)$, the claims follow from elementary calculus.

**Note (*to Reviewer):** The theorem is stated for an asymptotic continuation because integer partitions are defined on $\mathbb{N}$; the operational use is to analyze trends and then interpret discrete differences $\Delta \log\, M_{\text{comp}}(n)$ on $\mathbb{N}$.

## 4. Analysis: asymptotics, bounds, and a consciousness criterion

### 4.1. Asymptotic derivative and the efficiency gap

Hardy–Ramanujan asymptotics imply that $\log\, p(n)$ grows like $\pi\sqrt{2n/3} + O(\log\, n)$, hence its derivative scales like $O(n^{-1/2})$. Using this leading behavior, one obtains the approximation

$$\frac{d}{dn} \log\, p(n) \approx \frac{\pi}{\sqrt{6n}} - \frac{1}{n},$$

which is positive for large $n$ but decays to 0 as $n \to \infty$.

Thus the sign of $\eta(n)$ becomes a contest between a slowly decaying combinatorial gain and the marginal cost $\frac{d}{dn} F_{\text{eff}}(n)$, which depends on how difficult it is—under active inference—to maintain low surprisal while coordinating additional degrees of freedom.[2][1]

### 4.2. A minimal phenomenological approach for $F_{\text{eff}}(n)$

To make the critical-regime discussion concrete without committing to a specific biophysical micro-model, consider the approach

$$F_{\text{eff}}(n) = \alpha n + \beta n \log\, n + \gamma \sqrt{n},$$

where $\alpha, \beta, \gamma \geq 0$. The terms can be interpreted as: a linear per-unit maintenance cost, an information-theoretic coding/coordination cost growing like $n\log n$, and a cross-scale coordination barrier scaling like $\sqrt{n}$.

Under this approach,

$$\eta(n) \approx \left(\frac{\pi}{\sqrt{6n}} - \frac{1}{n}\right) - \left(\alpha + \beta(\log n + 1) + \frac{\gamma}{2\sqrt{n}}\right).$$

Critical points satisfy $\eta(n) = 0$, defining an implicit critical complexity $n_c$ where marginal gains balance marginal costs.

## 4.3. Rigorous growth bounds

Because $F_{\text{eff}}(n) \geq 0$ in the standard free-energy-bound reading, $M_{\text{comp}}(n)$ is always bounded above by $\Psi p(n)$.

**Theorem 2 (Upper bound)**
If $F_{\text{eff}}(n) \geq 0$, then

$$M_{\text{comp}}(n) \leq \Psi\, p(n).$$

Moreover, by Hardy–Ramanujan growth,

$$M_{\text{comp}}(n) = O\left(\exp\left(C\sqrt{n}\right)\right)$$

for an explicit constant $C > 0$ inherited from the partition asymptotic.

*Proof:* Immediate from the definition and the Hardy–Ramanujan asymptotic upper control.

**Theorem 3 (Lower bound under efficient adaptation)**
Assume $F_{\text{eff}}(n) \leq C\log n$ for all sufficiently large $n$. Then for large $n$,

$$M_{\text{comp}}(n) \geq \Psi\, n^{-C}\, p(n),$$

so $M_{\text{comp}}(n)$ still exhibits superpolynomial growth driven by $p(n)$.[3]

*Proof:* Substitute the assumption into the definition and apply Hardy–Ramanujan lower control on $p(n)$.

**Corollary 1 (Cost growth needed for saturation)**
If one requires $M_{\text{comp}}(n)$ to remain bounded as $n \to \infty$, then $F_{\text{eff}}(n)$ must grow at least on the order of $\sqrt{n}$ (up to logarithmic factors), matching the leading growth of $\log p(n)$.

## 4.4. Exact computation via Rademacher truncation (implementability)

Rademacher's exact formula expresses $p(n)$ as an absolutely convergent series, allowing numerical approximation by truncation to finitely many terms with controllable error. This makes $M_{\text{comp}}(n)$ computationally tractable even when one does not rely solely on the leading Hardy–Ramanujan term, provided $F_{\text{eff}}(n)$ can be estimated from a model or data stream.

## 4.5. An operational criterion for computational consciousness

We define a mathematically checkable regime condition and then add a self-modeling condition inspired by active inference.

**Definition 2 (Computationally efficient regime)**

A system at complexity $n$ is in a computationally efficient regime if $\eta(n) > 0$, i.e.

$$\frac{d}{dn} \log p(n) > \frac{d}{dn} F_{\text{eff}}(n).$$

**Definition 3 (Computationally conscious regime)**

A system is in a computationally conscious regime if:

1.  $\eta(n) > 0$ holds on a nontrivial neighborhood (in discrete terms, $\Delta \log M_{\text{comp}}(n) > 0$ for several successive $n$), and

2.  its generative model explicitly represents beliefs about its own internal trajectories (self-modeling under active inference), which is a standard extension of Markov-blanket hierarchies ("blankets of blankets") to nested internal models.

This definition deliberately targets efficient self-referential prediction (a computational property) rather than claiming to derive phenomenology.

# 5. Implications for AI/AGI and discussion

This section outlines how the proposed notion of computational mass can be instantiated in artificial systems, how it relates to existing free-energy–based approaches in AI, and how it might inform future work on AGI-scale architectures.

## 5.1. Computational mass as an AI design lens

The definition

$$M_{\text{comp}}(n) = \Psi \, p(n) \exp\left(-F_{\text{eff}}(n)\right)$$

is substrate-agnostic: it applies to any system for which we can (i) identify a discrete complexity index $n$, (ii) estimate a configuration-capacity proxy via $p(n)$, and (iii) define an effective free-energy–like quantity $F_{\text{eff}}(n)$ that summarizes predictive or generative performance under uncertainty. In artificial systems, these ingredients are often already present in disguised form—for

example, in the dimensionality of latent spaces, the number of active parameters, or the objective functions used for variational training.

Interpreted in this way, $M_{\text{comp}}(n)$ offers a compact scalar that trades off structural richness against adaptive cost. It can therefore serve as a design and diagnostic lens: architectures and training regimes that increase model size without reducing per-degree-of-freedom "free energy" (e.g., reconstruction error, variational loss, or prediction error) will exhibit saturating or even decreasing $M_{\text{comp}}(n)$, signaling diminishing returns from mere scaling.

## 5.2. Variational generative models as concrete examples

Many contemporary AI systems implement objectives that are variational in the same sense as FEP, even if they do not explicitly invoke Markov blankets. Variational autoencoders (VAEs), diffusion models, and other deep generative architectures train by minimizing divergences between approximate posteriors and target distributions, often via evidence lower bounds (ELBOs) or related free-energy–like functionals.

A simple instantiation is to consider a family of generative models indexed by latent dimension $d$, and identify $n = d$. For each model, define an effective free energy $F_{\text{eff}}(d)$ as a normalized negative average ELBO (or another suitable free-energy surrogate) on a validation set. The resulting

$$M_{\text{comp}}^{\text{gen}}(d) = \Psi \, p(d) \exp\left(-F_{\text{eff}}(d)\right)$$

then quantifies how much structural capacity (through $p(d)$) is actually converted into predictive or generative efficiency (through $\exp\left(-F_{\text{eff}}\right)$). Empirically, one could study how $M_{\text{comp}}^{\text{gen}}(d)$ scales with $d$, looking for regimes where its growth slows or reverses, indicating that additional latent dimensions no longer improve—and may even harm—overall computational mass.

More sophisticated versions could let $n$ measure effective intrinsic dimensionality, rank, or number of functional modules rather than raw parameter count, thereby aligning more closely with the idea of "effective degrees of freedom" within a Markov-blanketed architecture.

## 5.3. Active-inference agents and Markov-blanket hierarchies

Active-inference formulations of artificial agents explicitly mirror the structure assumed in the theoretical definition of computational mass: generative models over hidden states and observations, approximate posteriors, and action policies that aim to minimize expected future free energy. In such agents, the complexity index $n$ can naturally be associated with:

- the number of hidden-state dimensions;
- the number of distinct policies or policy hierarchies;
- or the number of nested Markov blankets in a multi-scale control architecture.

For each choice of $n$, one can, at least in principle, simulate agent–environment interactions, estimate $F_{\text{eff}}(n)$ from free-energy trajectories, and combine this with $p(n)$ to obtain $M_{\text{comp}}(n)$. Monitoring how $M_{\text{comp}}(n)$ changes as one refines the agent's generative model, increases policy depth, or adds hierarchical levels could reveal "sweet spots" where additional structure genuinely improves self-predictive, self-stabilizing behavior, versus regimes where free-energy costs outpace combinatorial benefits.

At higher levels of organization, Markov-blanket hierarchies allow entire agent collectives or socio-technical systems to be treated as single active-inference entities. In that case, $n$ might index effective roles, institutions, or communication channels, and $M_{\text{comp}}(n)$ would become a measure of collective computational mass. This opens a route to studying how organizational complexity and coordination costs shape emergent intelligence in multi-agent systems.

## 5.4. Towards AGI: critical regimes and self-modeling

The notion of a computationally conscious regime—where $M_{\text{comp}}(n)$ grows with $n$ and the system's generative model includes explicit self-modeling—provides a potential bridge between scaling laws for AI models and qualitative transitions in behavior. In practice, one could look for:

- complexity levels where increasing $n$ begins to yield super-linear gains in long-horizon predictive performance, robustness, or self-consistency;

- concomitant reductions in per-degree-of-freedom free energy, indicating better use of additional capacity;

- and the emergence of internal representations that encode not only external variables but also the model's own future states, decisions, or errors.

Such signatures would not amount to a proof of "AGI" or "consciousness," but they would mark a transition toward architectures that satisfy the operational conditions of a computationally conscious regime as defined in this work. This could guide the exploration of model families and training protocols, helping to distinguish mere parameter scaling from genuinely more powerful and self-referential forms of intelligence.

## 5.5. Relation to other mathematical theories of cognition

The proposed framework complements existing mathematical theories of cognition and consciousness rather than replacing them. For example, integrated information theory defines a quantity $\Phi$ intended to capture how much information is both differentiated and integrated within a system. Other work has proposed category-theoretic, dynamical-systems, or logical frameworks for natural intelligence. In this landscape, computational mass offers:

- a partition-theoretic view of structural capacity;

- an explicitly variational (FEP-like) measure of adaptive fit;

- and a single scalar synthesizing these into an "effective capacity for structured prediction."

One can imagine future work relating $M_{\text{comp}}(n)$ to $\Phi$ or to other measures, exploring when they agree, when they diverge, and what each reveals about the interplay between configuration space, dynamics, and information integration.

### 5.6. Summary of implications

In summary, *the concept of computational mass* suggests several concrete directions for AI/AGI research:

#### Theory

Use the analytical control provided by partition asymptotics and variational free-energy theory to derive scaling relations and critical conditions for self-modeling and robust intelligence.

#### Design

Use $M_{\text{comp}}(n)$ as a target or constraint when choosing model size and architecture, favoring regimes where added complexity still yields net gains in computational mass.

#### Diagnosis

Track $M_{\text{comp}}(n)$ across training or architecture changes to detect saturation points and redundant complexity.

These implications remain programmatic, but they provide a mathematically explicit agenda for linking physical resource constraints, combinatorial structure, and adaptive computation in both natural and artificial intelligent systems.

## 6. Methods: numerical estimation

This section outlines how to (i) estimate the effective free energy $F_{\text{eff}}(n)$ from active-inference–style simulations and (ii) compute the partition function $p(n)$ in practice using Hardy–Ramanujan asymptotics and truncated Rademacher series. The goal is to make the abstract construct

$$M_{\text{comp}}(n) = \Psi \, p(n) \exp\left(-F_{\text{eff}}(n)\right)$$

numerically accessible for concrete models.

### 6.1 Estimating $F_{\text{eff}}(n)$ from time series

Consider a family of active-inference agents indexed by complexity $n$ (for example, by the number of latent state dimensions, functional units, or policy branches). For each $n$, we simulate the agent interacting with an environment over a finite horizon and extract a time series of instantaneous variational free energy.

1. **Simulation setup and free-energy trajectory**

   For a fixed complexity level $n$, simulate the agent over a time interval $[0, T_n]$. At discrete times $t_k = k\Delta t$, $k = 0, \ldots, K_n$ with $T_n = K_n \Delta t$, compute the instantaneous (dimensionless) variational free energy

   $$F_n(t_k) = \mathbb{E}_{q_t}[\log q_t(\mu)] - \mathbb{E}_{q_t}[\log p(\mu, o_{t_k})],$$

   where $q_t(\mu)$ is the approximate posterior over internal states $\mu$ at time $t_k$, and $p(\mu, o_{t_k})$ is the generative model over internal and observed states. This is the standard free-energy expression used in FEP-based models of perception and action.

2. **Temporal averaging (ergodic estimate)**

   We approximate the time-averaged free energy at complexity $n$ by the discrete average

   $$\bar{F}_n := \frac{1}{K_n + 1} \sum_{k=0}^{K_n} F_n(t_k).$$

   Under an ergodicity assumption for the joint agent–environment dynamics, $\bar{F}_n$ converges (in probability) to the stationary average free energy associated with complexity $n$.

3. **Normalization across complexity levels**

   To compare systems with different $n$, we define a normalized effective free energy

   $$F_{\text{eff}}(n) := \frac{\bar{F}_n}{Z_n},$$

   where the scale factor $Z_n > 0$ is chosen according to modeling goals:

   *Per-degree-of-freedom normalization*: $Z_n = n$, so $F_{\text{eff}}(n)$ represents average free energy per effective unit.

   *Reference-based normalization*: select a baseline complexity $n_{\text{ref}}$ and choose $Z_n$ such that $F_{\text{eff}}(n_{\text{ref}}) = 0$. In this case, $F_{\text{eff}}(n)$ can be interpreted as an "excess free-energy cost" relative to the reference system.

4. **Numerical derivatives with respect to $n$**

   For a set of complexity values $\{n_i\}$, we estimate the derivative $\partial F_{\text{eff}}/\partial n$ using finite differences, e.g.

   $$\frac{\partial F_{\text{eff}}}{\partial n}\bigg|_{n \approx n_i} \approx \frac{F_{\text{eff}}(n_{i+1}) - F_{\text{eff}}(n_{i-1})}{n_{i+1} - n_{i-1}}.$$

   This derivative enters directly into the combinatorial efficiency function $\eta(n) = \frac{d}{dn}\log p(n) - \frac{d}{dn}F_{\text{eff}}(n)$, which determines whether computational mass increases or decreases with added

complexity. To mitigate noise, one may apply smoothing over $n$ (for example, local polynomial regression) before differentiating.

5. **Implementation remarks**
   In practice, the agent's variational update scheme determines how $q_t(\mu)$ and $F_n(t_k)$ are computed at each step. Many existing FEP and active-inference implementations already output approximate free-energy values at each update; our procedure treats those values as a time series from which $\bar{F}_n$ and $F_{\text{eff}}(n)$ can be extracted.

## 6.2 Computing $p(n)$: Hardy–Ramanujan and truncated Rademacher

The second ingredient in $M_{\text{comp}}(n)$ is the integer partition function $p(n)$, which we use as a proxy for the number of admissible internal configuration types at complexity $n$. For numerical work, one can use either asymptotic approximations or truncated exact series, depending on the required precision.

1. **Hardy–Ramanujan asymptotic approximation**
   For moderate to large $n$ (e.g., $n \gtrsim 50$), the leading Hardy–Ramanujan term provides a simple and efficient approximation:

$$p_{\text{HR}}(n) \approx \frac{1}{4n\sqrt{3}} \exp\left(\pi\sqrt{\frac{2n}{3}}\right).$$

In logarithmic form,

$$\log p_{\text{HR}}(n) \approx -\log\left(4n\sqrt{3}\right) + \pi\sqrt{\frac{2n}{3}}.$$

This form is convenient for analytical work (e.g., deriving $\frac{d}{dn}\log p(n)$) and for exploratory numerical studies of how $\log M_{\text{comp}}(n)$ scales with $n$.

2. **Truncated Rademacher exact series**
   When higher accuracy is required (for example, when fitting model parameters to data), one can use Rademacher's exact, absolutely convergent series for $p(n)$. In schematic form, this can be written as

$$p(n) = \frac{1}{\pi\sqrt{2}} \sum_{k=1}^{\infty} A_k(n) \sqrt{k} \frac{d}{dn}\left[\frac{\sinh\left(\frac{\lambda}{k}\sqrt{n-\delta}\right)}{\sqrt{n-\delta}}\right],$$

where $\lambda$, $\delta$ are explicit constants and $A_k(n)$ are arithmetic coefficients (Kloosterman-type sums) determined by modular properties of the partition generating function. In numerical practice:

- The infinite sum is truncated at some finite $K$, so only terms $k = 1, \ldots, K$ are evaluated.

- The truncation level $K$ can be chosen to balance accuracy and cost; for many $n$, relatively small $K$ already yields machine-precision approximations.

3. **Choice of approximation regime**

The appropriate method depends on the intended use:

For qualitative scaling analyses, phase-diagram construction, and analytic arguments about monotonicity and critical regimes, the Hardy–Ramanujan approximation is typically sufficient and much cheaper to evaluate.

For quantitative comparisons between $M_{\text{comp}}(n)$ and empirical or simulated $F_{\text{eff}}(n)$, a truncated Rademacher computation is preferable, as it eliminates approximation error from the partition side and isolates modeling assumptions in $F_{\text{eff}}(n)$.

4. **Combined numerical workflow**

A practical computational workflow is:

- **Step 1**

Use the Hardy–Ramanujan approximation to derive closed-form expressions for $\log p(n)$ and its derivative, and to identify approximate regions where the combinatorial gain $\frac{d}{dn} \log p(n)$ balances the marginal cost $\frac{d}{dn} F_{\text{eff}}(n)$.

- **Step 2**

Select a finite set of complexity values $\{n_i\}$ of particular interest (e.g., around candidate critical regimes) and recompute $p(n_i)$ using a truncated Rademacher series.

- **Step 3**

Recalculate $M_{\text{comp}}(n_i)$ with these refined $p(n_i)$ values and compare against the predictions made using the Hardy–Ramanujan approximation, thereby validating or adjusting the analytic conclusions.

Taken together, these procedures specify how to instantiate the abstract definition of computational mass in concrete models: $F_{\text{eff}}(n)$ is obtained from time-averaged variational free-energy trajectories in active-inference–like simulations, while $p(n)$ is computed using classical partition-theoretic methods. This makes the proposed quantity $M_{\text{comp}}(n)$ accessible to numerical exploration in both biological and artificial settings.

## Limitations

The present research framework is not a claim that partitions literally enumerate neural microstates, nor that $M_{\text{comp}}$ is directly measurable in biology without additional modeling layers. It is a

mathematically explicit proposal for coupling *configuration growth* (via $p(n)$) with *adaptive selection* (via $\exp(-F_{\text{eff}})$) under the broad FEP umbrella.

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
