# OpenReview forum: "Computational Mass of Living Matter: A Free‑Energy and Partition‑Theoretic Model of Consciousness"
_mathai.club/MathAI/2026/Conference — MathAI 2026 Conference Submission_

### Official Review · Reviewer_hBaD · 2026-03-10
**Ambitious theoretical proposal with limited empirical grounding**

**Rating:** 3
**Confidence:** 4

**Review:**

# Review

## Summary

The paper proposes a theoretical construct called **computational mass**, defined as

$$
M_{\text{comp}}(n)=\Psi,p(n)\exp(-F_{\text{eff}}(n)).
$$

Here (p(n)) denotes the integer partition function and (F_{\text{eff}}(n)) is a normalized effective variational free energy derived from the free energy principle (FEP). The proposed quantity aims to capture the trade-off between combinatorial internal configuration capacity and adaptive inference efficiency.

The framework introduces a dimensionless complexity index (n) derived from an energy scale

$$
n=\left\lfloor \frac{E_{\text{tot}}}{\epsilon_0} \right\rfloor ,
$$

and studies the behavior of (M_{\text{comp}}(n)) using asymptotic properties of the partition function, including the Hardy–Ramanujan formula.

The paper further defines a **computationally conscious regime** using the condition

$$
\frac{d}{dn}\log p(n) > \frac{d}{dn}F_{\text{eff}}(n),
$$

which indicates that combinatorial gains exceed marginal free-energy costs.

---

# Strengths

### 1. Ambitious interdisciplinary scope

The paper attempts to connect concepts from:

* the free energy principle
* analytic number theory
* theoretical neuroscience
* AI scaling laws

This interdisciplinary approach is intellectually interesting and could stimulate further discussion about mathematical measures of adaptive computational capacity.

---

### 2. Explicit mathematical formulation

The proposed framework introduces a well-defined scalar quantity and provides analytical results including monotonicity conditions and asymptotic analysis. For example, the growth of the partition function is approximated using

$$
\log p(n) \approx \pi\sqrt{\frac{2n}{3}}.
$$

This allows the authors to derive approximate expressions such as

$$
\frac{d}{dn}\log p(n) \approx \frac{\pi}{\sqrt{6n}} - \frac{1}{n}.
$$

These derivations provide some mathematical structure to the proposed measure.

---

# Major Weaknesses

## 1. Weak justification for the partition-based complexity model

The central modeling step is the use of the integer partition function (p(n)) as a proxy for the number of admissible internal configurations.

However, the connection between integer partitions and the configuration space of biological or artificial systems is not convincingly justified. Real neural or computational systems do not naturally correspond to unlabeled integer partitions.

Alternative complexity measures such as entropy, graph complexity, or state-space dimensionality might be equally plausible. The choice of partitions therefore appears somewhat arbitrary.

---

## 2. Lack of empirical validation

The framework is entirely theoretical. The paper outlines a numerical workflow but does not present:

* simulations
* empirical experiments
* case studies on AI models
* biological data analysis

Without empirical validation, it is difficult to assess whether the proposed quantity provides meaningful insight.

---

## 3. Key quantities are underspecified

Several components of the framework remain operationally unclear:

* how $(F_{\text{eff}}(n))$ should be estimated in practice
* how the energy resolution parameter (\epsilon_0) should be selected
* how the normalization constant (\Psi) should be chosen

Because these quantities directly determine the value of (M_{\text{comp}}(n)), clearer definitions are necessary for practical use.

---

## 4. Claims about consciousness are speculative

The paper introduces the notion of a **computationally conscious regime** defined by

$$
\frac{d}{dn}\log p(n) > \frac{d}{dn}F_{\text{eff}}(n).
$$

However, the connection between this mathematical condition and actual consciousness is not convincingly established. The definition appears largely terminological rather than grounded in empirical neuroscience or cognitive theory.

---

## 5. Limited engagement with related literature

The paper would benefit from a deeper comparison with existing frameworks such as:

* integrated information theory
* predictive processing and Bayesian brain models
* scaling laws in machine learning

Currently the discussion of related work remains relatively brief.

---

# Questions for the Authors

1. Why is the integer partition function an appropriate proxy for configuration capacity in biological or artificial systems?

2. Can the authors provide a concrete example where (M_{\text{comp}}(n)) is computed for a real or simulated system?

3. How sensitive are the results to the choice of parameters such as (\epsilon_0) and the normalization scheme?

4. What empirical predictions does the proposed framework make that distinguish it from existing theories?

---

# Overall Assessment

The paper presents an interesting mathematical proposal linking combinatorial configuration growth with free-energy-based adaptive inference. However, the modeling assumptions are weakly justified and the framework lacks empirical validation or concrete demonstrations.

At its current stage, the work reads more like a speculative theoretical proposal than a mature scientific contribution.

---

# Recommendation

Score: **3 / 10 (Reject)**

Confidence: **4 / 5**

While the mathematical exposition is clear, the conceptual foundations and practical relevance require substantial strengthening before the work would be suitable for publication.

---

### Official Review · Reviewer_fsTi · 2026-03-12
**The paper introduces "Computational Mass" (Mcomp), a dimensionless construct designed to quantify the complexity of living or artificial systems**

**Rating:** 7
**Confidence:** 4

**Review:**

Summary: The paper introduces "Computational Mass" (Mcomp), a dimensionless construct designed to quantify the complexity of living or artificial systems. The author defines a complexity index n and uses the integer partition function p(n) as a proxy for structural capacity, coupled with an effective free energy term Feff(n) derived from the Free Energy Principle. The paper proves monotonicity properties and bounds for this metric, identifying "computationally efficient" and "conscious" regimes based on the trade-off between combinatorial capacity and energetic cost. It concludes by mapping this framework onto variational AI models (VAEs, diffusion models) to offer a theoretical basis for analyzing scaling laws and AGI.

Strengths: Interdisciplinary Synthesis: The paper successfully bridges analytic number theory (partition theory) and variational Bayesian methods (FEP), offering a unique theoretical tool.
Formal Rigor: Unlike many papers on "consciousness" or "complexity," this work provides rigorous definitions, lemmas, and theorems (e.g., Theorem 1 on monotonicity) to support its claims.
Actionable Framework: The discussion on applying Mcomp to variational generative models (Section 5.2) provides a concrete path for empirical validation in AI research, moving beyond pure speculation.
Generalizability: The framework is substrate-agnostic, applying equally well to biological systems (neuroscience) and artificial architectures (AGI), fitting the MathAI scope perfectly.

Weaknesses: Proxy Validity: The assumption that integer partitions p(n) accurately model "internal configurations" is a theoretical proxy. While mathematically tractable, the link to actual biological or neural states is abstract. The paper acknowledges this limitation but the validity of the result hinges on accepting this abstraction.
Empirical Gap: The paper is purely theoretical. While it suggests how to estimate Feff and p(n) numerically, it lacks even a toy experiment or simulation to demonstrate the metric's behavior on a simple AI model (e.g., a small VAE).

Recommendation: Accept.
This paper represents the kind of deep theoretical contribution that MathAI should champion. It connects advanced mathematical domains to fundamental AI problems with rigor and originality. Despite the abstract nature of the biological mapping, the mathematical construction is sound, novel, and highly relevant to the conference's scope.

---

### Decision · Program_Chairs · 2026-03-20

**Decision:**

Accept (Poster)

**Comment:**

Dear Author(s),

On behalf of the Program Committee of the International Conference on Mathematics of Artificial Intelligence (MathAI 2026), we are pleased to inform you that your paper has been accepted for a poster presentation at MathAI 2026.

Your paper was evaluated through a rigorous two-stage review process involving both automated screening and expert review by members of the Program Committee. The reviewers recognized the quality and contribution of your work.

Important Note: The reviewers have recommended final revisions to your manuscript before the conference. Please ensure that all reviewer comments are carefully addressed in your camera-ready version. We trust that you will complete these revisions before the conference deadlines.

Presentation details:

    Format: Poster presentation

    Mode: You may present either in person (offline) at the conference venue in Sirius, Russia, or remotely via Zoom. Please indicate your preferred mode when confirming your participation.

    Conference dates: March 30 - April 3, 2026

    Website: https://mathai.club

Next steps:

    Please confirm your participation and presentation mode by replying to this email (mathai.club@yandex.ru) no later than March 15, 2026 18:00 Moscow time.

    If you plan to attend in person, the organizing committee will provide accommodation details separately.

    Please prepare your final camera-ready manuscript according to the formatting guidelines available at https://mathai.club and upload it to OpenReview by March 15, 2026 18:00 Moscow time. Ensure that all reviewer feedback has been incorporated into this final version.

Should you have any questions regarding the program, logistics, or your presentation, please do not hesitate to contact us.

We look forward to your contribution to MathAI 2026.

With kind regards,

MathAI 2026 Program Committee
International Conference on Mathematics of Artificial Intelligence
https://mathai.club
OpenReview: https://openreview.net/group?id=mathai.club/MathAI/2026/Conference
Telegram: https://t.me/MathAI_club
Email: mathai.club@yandex.ru